# The Effect of Maternal Exposure to a Diet High in Fats and Cholesterol on the Placental Function and Phenotype of the Offspring in a Rabbit Model: A Summary Review of About 15 Years of Research

**DOI:** 10.3390/ijms241914547

**Published:** 2023-09-26

**Authors:** Delphine Rousseau-Ralliard, Pascale Chavatte-Palmer, Anne Couturier-Tarrade

**Affiliations:** 1BREED, INRAE, UVSQ, Université Paris-Saclay, 78350 Jouy-en-Josas, France; delphine.rousseau@inrae.fr (D.R.-R.); pascale.chavatte-palmer@inrae.fr (P.C.-P.); 2BREED, Ecole Nationale Vétérinaire d’Alfort, 94700 Maisons-Alfort, France

**Keywords:** DOHaD, nutrition, placenta, sex effects, phenotype, offspring

## Abstract

The rates of obesity and being overweight are increasing all around the world, especially among women of childbearing age, in part due to overconsumption of lipids. The aim of this summary review was to present the cellular and molecular effects of a hyperlipidic high-cholesterol (H) diet on the maternal and offspring phenotype at the early embryonic, neonatal, weaning and adult stages while considering the effects of sex and to identify the window(s) of vulnerability linked to this exposure in a rabbit model. Before breeding, the H diet induced dyslipidemia and aortic atherosclerosis lesions and increased the number of atretic follicles. In the offspring, the H diet disrupted the embryonic phenotype and induced fetal hypotrophy associated with sex-specific disturbances of the feto-placental unit. In adulthood, the offspring of the H dams were heavier and hyperphagic and had increased blood pressure associated with disturbed gonadal development in both sexes. Vulnerability windows were explored via embryo transfers. The maternal gestational diet was shown to play a key role in the feto-placental phenotype, and preconception programming was unquestionably also observed. These two periods could represent windows of intervention in the context of obesity or being overweight to limit fetal and placental consequences.

## 1. Introduction

The incidence of obesity and being overweight is rising in the United States, Europe and most other developed countries [1] due to dietary transitions and increased fat consumption. In Europe, fat consumption often exceeds recommended intakes, with maximum intakes ranging from 37% of total energy in the western countries to 46% in the most southern countries. In particular, the fat intake of women of childbearing age is 35.4% in the UK [2] and 39.6% in France, whereas recommendations suggest that it should be between 30 and 35% of the diet [3,4]. The risk of overconsumption of calories is therefore very real and follows obesity levels in Europe perfectly.

Even if the total fat intake is a source of concern, the quality of fat should also be considered. Western countries are faced with overconsumption of linoleic acid [5,6], a more important ω-6 polyunsaturated fatty acids (PUFA), through meat consumption, whose high linoleic acid (LA) content is particularly due to animal feed in intensive livestock farming (feed rich in soya and maize), the replacement of saturated fatty acids (SFA) with PUFA or through the consumption of processed foods prepared from cheap oils (soya, sunflower and maize oils) by the food industry, leading to an increase in the ratio of ω-6 PUFA to linoleic acid, sunflower and maize oils [5,7] and an increase in the ω-6/ω-3 PUFA ratio and weight gain [6,8,9].

This metabolic status is not without consequences, since it is associated with an increased risk of developing noncommunicable diseases in adulthood, such as type-2 diabetes, cardiovascular disease, hypertension and stroke, which are often associated with non-alcoholic fatty liver disease, which reduces both quality of life and life expectancy and is a leading cause of public health concern.

In France, the ObEpi survey (Roche) indicated in 2012 that 15.7% of French women were obese, 11.1% of whom were of childbearing age [10]. According to the latest results of the ObEpi 2 survey, almost half the French population (47%) is overweight or obese. Excess weight is known to reduce fertility and increase obstetric complications, such as miscarriage, gestational diabetes, hypertension and pre-eclampsia [11]. In addition, children born to obese or overweight mothers have increased adiposity and a higher risk of metabolic diseases in adulthood [12]. Thus, a maternal high-fat diet during pregnancy is associated with rapid weight gain and fetal fat mass increase at an early stage. Also, a high-fat diet during pregnancy can cause the activation of proinflammatory cytokines. In addition, maternal insulin resistance and inflammation lead to increased adipose tissue lipolysis, and increased free fatty acid intake during pregnancy (˃35% of energy from fat) causes a significant increase in free fatty acid levels in the fetus [13].

Pregnancy is therefore a period of vulnerability, as described in the concept of the developmental origins of health and disease (DOHaD) [14], which states that any disruption to the maternal environment (including diet) during the sensitive phases of development can have effects on feto-placental development, with long-term effects on the health of the offspring, as demonstrated by epidemiological studies, and significant associations between low birth weight and cardiovascular disease or type-2 diabetes in adulthood [15]. However, the increased risk of these noncommunicable diseases in the offspring depends on the stage of exposure during the prenatal period. In response to an altered in-utero environment, the fetus will develop adaptive responses to improve its survival. When the prenatal and postnatal environments differ, the risk of developing noncommunicable diseases in adulthood increases [16].

Currently, studies on humans usually focus on the effects of maternal overnutrition during the second or third trimester of pregnancy, when key processes such as organogenesis are already completed. Indeed, pregnant women are most often included in cohorts after pregnancy diagnosis at the end of the first trimester. Thus, the specific effects of maternal overfeeding before conception or in the periconceptional period are often underestimated [17], although works on experimental animals and a limited number of human studies clearly show that the body composition and nutrition during this period, known to affect fertility and early embryo viability [18,19,20,21,22], also induce long-term metabolic effects in offspring [23,24,25,26,27]. It is henceforth recognized that in modern westernized societies, there is high dietary consumption of foods rich in n-6 PUFA, especially linoleic acid and, to a lesser extent, arachidonic acid, which could have detrimental consequences for the fetus and neonate due to excessive exposure to these fatty acids (FA), despite their central role in the development of the central nervous system as part of the membrane structure and in the metabolism and signal transduction of cells [28].

The placenta is a temporary organ of fetal origin located at the fetal-maternal interface, ensuring nutritional, metabolic, endocrine and immune exchanges. Any disturbance during its development can alter its structure and functions, particularly the supply of nutrients from the mother to the fetus, which controls fetal growth and development [29]. An inadequate maternal nutritional environment can alter placental gene expression, vascular perfusion and endocrine function and induce inflammation [30,31,32]. The placenta is therefore considered to be a fetal programming agent (i.e., its response to the maternal environment will regulate fetal adaptative mechanisms) [33]. In addition, the placental response to an adverse environment often differs between males and females, leading to a sex-specific placental signature and, in turn, a contribution to sex-specific effects in offspring [29,34].

In the past 10–15 years, the authors’ laboratory has explored the effects of a maternal high-fat high-cholesterol in a rabbit model. The aim of this review is to summarize this comprehensive work, focusing on the cellular and molecular effects of a diet enriched with cholesterol (0.2%) and fats (ω-6/ω-3 ratio = 6.86) on the maternal and offspring phenotype at different stages (fetal, weaning and adult), considering sex-related effects and discriminating between effects due to preconception exposure to the experimental diet and those linked to gestational exposure.

## 2. The Rabbit Model

The studies described below were carried out using rabbits [34,35,36,37,38,39]. This model is the third most widely used model in animal experimentation in the European Union after rodents for studying human pathologies and for toxicology studies [40].

In terms of reproduction, it offers several advantages. Females are sexually mature at around 4–5 months. Their uteri are duplex compared with rodents, consisting of two long horns and two cervices so that two different groups of embryos can be surgically transferred to the same recipient female without risk of confusing the groups. Moreover, the rabbit’s reproductive cycle is short, being 29–31 days (versus 19–21 and 21–23 days in mice and rats, respectively), facilitating easy transgenerational studies.

The rabbit is one of the few species in which ovulation is induced by mating, making it possible to accurately date the embryonic age in hours (h) or days (d) post-coitum (p.c.). Fertilization occurs at 10 h p.c., and the second polar globule becomes visible at around 14 h p.c. Until the 16-cell stage, blastomeres divide every 7–8 h. The morula stage is reached at 60 h p.c. and the blastocyst stage at 72 h p.c. The blastocyst stage is characterized by the appearance of the blastocoel cavity and the differentiation of the inner cell mass (ICM or embryoblast) and the trophoblast. At 4 d.p.c., the size of the blastocyst reaches 1 mm. It grows to 6 mm in diameter until implantation at day 6 and 18 h, closer to that of the human species, whereas implantation occurs on days 4 and 5 in mice and rats, respectively [41]. This size allows sufficient material to be collected for morphological and molecular analyses. Embryonic microdissection makes it possible to separate the ICM from the trophectoderm and perform physiological analyses, and the embryo can be visualized with micro-ultrasonography [42].

Embryonic genome activation (EGA) in rabbits extends over several cell cycles as in most mammals, including humans, but unlike mice. These interspecies differences are significant for the regulation of EGA. The rabbit genome is fully active from the 8- to 16-cell stage. Passive DNA demethylation occurs between the 4- and 16-cell stages, followed by de novo remethylation at the same stage.

The next stage is gastrulation and corresponds to a key moment in blastocyst differentiation when the rabbit embryo is discoid in shape, with two cell layers (the epiblast and hypoblast). This differentiation begins on day 6 p.c. just before implantation [40]. Chorioallantoic mesometrial placentation is established on day 8 p.c.

The rabbit placenta is discoid and hemochorial like that of humans and rodents. As for rodents, the exchange zone between maternal and fetal blood is called the labyrinthine zone. The hemodichorial structure of the rabbit placenta, however, with chorionic two-cell layers separating maternal and fetal blood, is closer to that of the human placenta in the last trimester (hemomonochorial placenta) than that of the rodent placenta (hemotrichorial placenta) (i.e., three trophoblastic layers) [43]. The placenta grows rapidly during the first half of gestation, followed by an increase in fetal weight due to an increase in fetal-maternal exchanges and the placental exchange surface area. Placental giant cells that originate from the trophoblast are present in the rabbit placenta from day 10 p.c. but not in the human placenta. They are involved in the remodeling of the spiral arteries [44], where they replace the endothelial cells as demonstrated in rodents, resembling the invasive trophoblast in humans.

In terms of hemodynamic parameters, gestation is characterized by an increase in maternal blood pressure throughout gestation, emphasizing the relevance of this model for the study of pathologies that affect maternal arterial pressure or placental perfusion. Finally, the size of the rabbit allows the monitoring of feto-placental growth at several stages of gestation without anesthesia unlike rodents, using standard ultrasound equipment also used in human medicine [45,46,47].

In conclusion, due to the anatomical and physiological characteristics of the rabbit described above, more relevant for transposition to humans and for our scientific questions than those of rodents, the effects of a diet high in cholesterol and fats were evaluated using this model.

## 3. The High-Cholesterol and High-Fat Diet (H Diet)

Exposure of the female rabbits to a diet enriched with cholesterol and fat (H diet) is aimed to induce hyperlipidemia, hypercholesterolemia and atherosclerosis without inducing obesity nor abortion in pregnant females. A diet including 0.2% cholesterol and 8% fat was used, similar to that described by Napoli et al. [48] in their seminal work. Preliminary experiments with this diet in a laboratory confirmed that it was compatible with neonatal viability. Soy oil, used as sources of lipids, and cholesterol were used to prepare the H diet. Qualitatively, soy oil contains linoleic acid (51%), oleic acid (23%) and alpha-linolenic acid (8%). It is therefore an important natural source of ω-6 and ω-3 PUFA. Its ω-6/ω-3 ratio is 6.86 which, according to the French Food Safety Agency, is too high, as the ideal ratio should be 5. The rest of the diet was strictly identical to that administered to the controls so that the H diet provided 16% more energy than the control diet (C). New Zealand rabbits were fed either the H diet or C diet from 10 weeks of age (pre-pubertal) in the form of pellets, and the diets were fed ad libitum [34,35,36,37,38,39].

## 4. Effects of the H Diet on the Phenotype of Female Does (F0)

The effects of the diet were assessed before the does were bred (i.e., at 18 weeks of age and after 8 weeks of exposure to the H diet (Figure 1)). Weight gain was not significantly different between the H and C females [35,36], although the average weekly energy intake during the 10–17-week period was higher in the H does compared with the C does. This 2-month exposure did not affect the bodyweight nor organ biometry, but the H does had a significantly higher body fat content compared with the C group. The absence of excess weight in the fatty H females at 18 weeks of age could be explained by adaptations of food intake to their energy needs. In fact, food intake was reduced in H females compared with C females, whereas the average weekly energy intake was higher in the H does compared with the C does. The plasma concentrations of total cholesterol, HDL and triglycerides, however, were significantly higher in the H group. In addition, atherosclerotic lesions were observed in the aortas of all H females.

In terms of reproduction, the luteinizing hormone (LH) surge in response to the challenge of gonadotropin-releasing hormone (GnRH) was significantly higher in the treated females compared with the controls at 13 weeks of age, suggesting a potential precocious puberty. An early onset of puberty and a high LH pulse frequency has also been demonstrated in obese rats [49]. A fat-related signal was shown to facilitate the activation of hypothalamic GnRH release and advance the onset of puberty [49]. At 18 weeks of age, no difference in LH was found anymore between the two groups nor leptin or progesterone, whereas estradiol was significantly lower in the H group compared with the C group. Decreased circulating estradiol concentrations were also observed in an obese rat model [50]. Furthermore, the anti-apoptotic role of estradiol could explain the increase in follicular atresia [51] as well as an effect of plasma cholesterol, since a follicular atresia has been reported in a rat model fed with a high-cholesterol diet [52]. Concomitantly, the number of atretic follicles was significantly increased, and the number of antral follicles was significantly reduced in H females compared with the C group [36] (Figure 1). At this stage, the females were bred with a control male, and dietary exposure to H or C was continued throughout gestation. At mid-gestation, the plasma concentrations of total and HDL cholesterol were still significantly elevated in the H group and associated with a significant increase in fasting plasma insulin concentrations in the H group.

At the end of gestation (i.e., 28 d.p.c. (gestation lasts 30–31 days in rabbits)), the maternal plasma total cholesterol and FA concentrations (H > C, *p* < 0.01) were still significantly higher in the H females than the C females. Maternal plasma FA profiling reflected dietary FA intake, with an increase in the proportion of ω-6 PUFA at the expense of MUFA in the H females compared with the C females [34].

## 5. Effects of the H Diet on the Embryo

In order to assess the effects of the H diet on the embryonic phenotype at the 16–20 cell stage (EGA stage), F0 females were superovulated for 3 days prior to natural mating. Embryos at the 16–20 cell stage were recovered from oviducts rinsed with PBS at 48 h p.c. immediately after euthanasia. Transcriptomic analyses were performed on whole embryos. Differential microarray screening showed that adipophilin was significantly overexpressed in the H embryos compared with the C embryos [35] (Figure 1). This gene codes for a protein involved in the formation of lipid droplets and intracellular lipid storage [53,54].

At pre-implantation (blastocyst stage), immunodetection of adipophilin associated with a lipid marker “Nile Red” revealed the accumulation of lipid droplets in the cytoplasm of trophoblastic cells in H compared with C blastocysts [34] (Figure 1), prompting us to evaluate the expression of other genes involved in metabolism. The relative expression of the glucose transporter genes solute carrier family 2 (*SLC2A1*) and *SLC2A3* was significantly increased in the H blastocysts, whereas the transcripts of the neutral amino acid transporter *SLC38A2* were significantly decreased in the H blastocysts. Expression of the *SLC38A1* gene, coding for another neutral amino acid transporter, remained unchanged. Transcripts of the nuclear receptor liver X receptor α (*LXR-α*) and *adipophilin* were significantly reduced in the H blastocysts compared with the C blastocysts. Gene expression of LXR-α, known to regulate ATP binding cassette transporter G1 (ABC-G1), was indeed significantly higher in the H blastocysts than in the C blastocysts, while the gene expression of ABC-A1, Fatty acid translocase (CD-36), LDL receptor (LDL-R) and the nuclear receptor PPARγ remained unchanged.

The maternal H diet affected the trophoblast at the blastocyst stage, characterized by an increase in lipid contents and a deregulation of gene expression in the blastocyst, as observed in bovine embryos, which are sensitive to maternal hyperlipidemia [55]. The difference in *adipophilin* gene expression observed between the 16–20 cell and blastocyst stages could be due to a rapid regulatory loop aimed at limiting the accumulation of droplets in the embryo, as it is known that a decrease in adipophilin reduces the formation of lipid droplets [56]. Furthermore, it could also be an adaptive response to the maternal H environment related to the contents of the uterine fluid, because the H environment differed from the C group [57], or to an alteration of follicular fluid contents in the oocyte, as evidenced in humans [58] and cattle [59].

## 6. Effects of the H Diet on Feto-Placental Development

The effects of the H maternal diet on feto-placental development were assessed by ultrasound during gestation. From day 9 p.c., the sizes of the H embryonic vesicles were significantly reduced compared with the controls. At 14 d.p.c., the H fetuses’ craniocaudal length was significantly shorter than that of the C fetuses. At 21 d.p.c., no difference was observed. In contrast, at 28 d.p.c. (gestation lasts 30–31 days in rabbits), the length and thickness of the placenta, measured by ultrasound, were greater in the H group versus the C group, with a reduced fetal abdominal perimeter in the H group having no effect on the fetal heart rate [35].

At 28 d.p.c., the feto-placental units were collected by caesarean section. The litter size was not statistically different between the groups, indicating that the H diet did not affect implantation. The H fetuses were hypotrophic (−17.8% in weight compared with the controls), and the placental weight was reduced in the H group, but the fetal weight/placenta weight ratio remained unchanged, indicating that placental efficiency was maintained through placental adaptation. The liver and kidney relative weight to body weight were unchanged. No sex-specific differences were observed for these measurements [35] (Figure 1 and Figure 2).

In terms of biochemistry, the H fetuses were characterized by sexual dimorphism, with greater fetal dyslipidemia in males than in females. Indeed, at the end of gestation, the fetal biochemical phenotype was characterized by significantly higher plasma concentrations of total cholesterol and triacylglycerol in the H fetuses compared with the C fetuses. This increase in cholesterol was due to increases in the male fetuses as the female fetuses were not affected. Plasma total fat concentrations were also significantly higher in the H fetuses, mainly due to increased SFA and MUFA in the H males compared with the C males but not in females. Conversely, increased plasma PUFA concentrations were observed in both H males and females (Table 1). An elevated intake of n-6 PUFA, specifically LA, during pregnancy could harm the placenta and the development of other fetal organs such as the fat tissue, liver and cardiovascular system [28].

These differences in circulating lipids between H males and females could be the result of altered fetal hepatic metabolism or lipid transfer from placenta to fetus. To answer this question, fetal hepatic lipid profiles were explored, but no significant difference in hepatic cholesterol, cholesterol ester or triacylglycerol concentrations were observed between groups. No hepatic membrane hypertrophy was observed, as the analysis of hepatic phospholipids, used as a proxy for membranes, indicated that the concentrations of phospholipid FA were similar between the H and C livers. Qualitatively, however, the FA profiles of the H livers were richer in ω-6 PUFA than the C livers, at the expense of other classes of SFA and MUFA. On the other hand, increased concentrations of lipids stored in the form of triglycerides within the liver tissue (neutral lipids) were observed in both H fetuses (*p* < 0.01), regardless of sex, and their dams. In terms of FA in the neutral lipid fraction, the concentrations of SFA and PUFA were significantly higher in the H livers, while those of MUFA remained unchanged.

Considering these results, the expression of hepatic genes involved in lipid and cholesterol metabolism was explored. The relative gene expression of the acetyl-CoA carboxylase (ACC-α), which is a limiting enzyme in the de novo synthesis of FA, and gene expression of LDL-R were reduced in the H livers compared with the C livers. Moreover, no changes in gene expression were observed for fatty acid synthase (FAS), which is involved in the synthesis of FA, fatty acid transport protein (FATP-4), adipophilin, peroxisome proliferator-activated receptor (PPAR-α), liver X receptor (LXR-α), sterol regulatory sterol element-binding protein (SREBP-2) or hydroxymethylglutaryl-coenzyme A reductase (HMG-coA reductase), all of which are involved in lipid metabolism. No sexual dimorphism was observed in our gene expression data [34].

As the hepatic results could not explain the plasma FA profiles of the H fetuses, the placenta, an essential organ in the supply of nutrients, was studied. At 28 d.p.c., structural analysis of the labyrinthine zone, involved in feto-maternal exchanges, showed an accumulation of lipid droplets in the H placentas compared with the C placentas. The placental lipid profiles showed an unchanged concentration of total cholesterol between the H and C placentas. However, the concentration of cholesterol esters was significantly higher in the H placentas than in the C placentas in both males and females. The concentrations of triacylglycerol were also significantly higher in the H placentas than in the C placentas but with a greater increase only in the H males than in the C males.

As for the liver, placental FA were analyzed by separating phospholipids (membranes) from neutral lipids (lipid storage). In the placenta, the phospholipids appeared highly saturated whatever the diet was (barrier role). The FA concentration of phospholipids increased only in females (sexual dimorphism) (Table 1), reflecting an increase in membrane mass whatever the diet was. The neutral lipids, consisting of triglycerides and cholesterol esters, were rich in SFA and low in PUFA in the C fetuses. The nature of the FA stored in the H fetuses was modified, with a rebalancing between the three classes of FA whatever the sex, and the placental concentration was higher in the H fetuses than in the controls. However, FA storage was also significantly higher in the female placentas than in the males, whatever the diet (Figure 2). At the end of gestation, the placentas presented a lipid droplet accumulation corroborating the high concentrations of cholesteryl esters, triacylglycerol and stored FA in the H placentas without any change in total cholesterol content. This result could be related to an increased uptake of maternal lipoproteins through the placenta, as the maternal plasma HDL concentrations were increased at mid-gestation in the H does and as described in female hamsters [60].

The fetal dyslipidemia and hypotrophy observed in the H fetuses, as well as the accumulation of lipid droplets in the placenta, strongly suggest an alteration in the placental gene expression involved in the transfer and metabolism of nutrients. Moreover, the maternal diet altered the placental membrane PUFA composition in the H females, which is likely to induce changes in membrane fluidity, affecting membrane protein functions such as ion channel, transporter and receptor function, and the maternal H diet could impair nutrient transfer [61]. For this reason, the expression of genes involved in placental transfer and already analyzed in blastocysts was explored. As glucose transporters were affected by the H diet at the early blastocyst stage [62], this increase could stimulate glucose uptake in the H embryos, leading to a delay in development as demonstrated by Ramin et al. [63]. This hypothesis was also supported by the IUGR observed when using ultrasound from the ninth day of pregnancy [35]. However, the expression of the glucose transporter genes *SLC2A1* and *SLC2A3* was not affected anymore in late gestation by the maternal H diet. In contrast, *SLC38A* was downregulated in both the H blastocysts and placentas. The expression of *SLC38A1*, involved in the transport of neutral amino acids, was significantly reduced in the H placentas compared with the C placentas, while those of the transporters *SLC38A2* and *SLC38A4* were unchanged. This downregulation could contribute to the IUGR phenotype, as a decrease in system A activity, involved in amino acid transfer, in the placenta in human and animal models has been associated with IUGR [64,65]. These data demonstrate that trophoblastic adaptive strategies in response to maternal hyperlipidemia start at an extremely early stage of development and that these adaptive mechanisms persist throughout gestation.

Concerning lipid metabolism and transport, expression of the *LDL-R*, *CD3*6 and *LXR-α* genes, involved in cholesterol exchange pathways, was significantly reduced in the H placentas compared with the C placentas. Expression of the *LXR-α* gene was significantly reduced in the placenta of H males but not in females. Expression of the cholesterol transporter *ABC-G1*, controlled in particular by *LXR-α*, was also reduced in the H group. However, the gene expression of ABC-A1, HMG-CoA reductase and SREBP-2, involved in cholesterol trafficking pathways, remained unchanged. No differences in gene expression were observed between the groups for genes involved in lipid metabolism, *adipophilin*, *FATP-4*, *PPAR-γ* and its partner *retinoid X receptor (RXR-α)*.

These results [34] demonstrate that the male and female feto-placental units were not metabolically equivalent and did not react the same way to disturbances in the maternal nutritional environment. Maternal FA profiling was also reflected in the fetal tissues (plasma and other organs), both qualitatively (FA profiling) and quantitatively, with higher concentrations in the F1 H fetuses and sexual dimorphism in response to the H diet. The F1 placentas from the H females retained FA delivered in excess by their mothers (F0), whereas the male F1 placentas from the H dams were unable to do so and were already dyslipidemic in-utero. The ultrastructural accumulation of lipids in the placentas at the end of gestation, confirmed by a lipidological approach, was partly explained by the placental decrease in gene expression involved in the transfer of cholesterol and triglycerides (Figure 2).

Specific sexually dimorphic changes in the placenta were observed in terms of gene expression and physiological adaptation, with a relative protection of the female fetuses from developing dyslipidemia, as demonstrated by increased lipid storage in the female placentas compared with the males, while the H males exhibited higher plasma FA and cholesterol concentrations compared with the females. More detailed knowledge on these sexually dimorphic adaptive mechanisms in response to a maternal high-fat diet will help us to explain various sex-specific pathologies at adulthood. Sprague Dawley rats were fed either standard chow or “cafeteria” diets across pre-pregnancy, gestation and lactation, and it was shown that the impact of maternal obesity on adiposity and liver gene expression was more marked in adult males, highlighting the sex-specific detrimental effects of maternal obesity on offspring [66]. Some of the differences between sexes may be linked to sex hormones, especially estrogen, which exerts protective effects on adipose tissue, skeletal muscle, liver and pancreatic β-cells. Thus, estrogen has been shown to protect against high-fat diet-induced insulin resistance and glucose intolerance in mice [67], and work involving estrogen receptor-α (ERα) knockout mice suggests that estrogen may improve hepatic insulin sensitivity through ER-α signaling [68].

Differences in placental growth and nutrient transfer may also contribute to sex-specific changes in offspring [69]. Among the key fatty acid synthesis pathway enzymes whose offspring’s hepatic expression changed in response to exposure to the maternal cafeteria diet (in rats), the only sex-specific effects were on PPARα expression, which was increased in males, and CPT1 expression, which decreased, but not in weaned females [66]. However, in rats as well, a maternal high-LA diet increased the circulating leptin in female offspring but not in males, decreased the circulating adiponectin in males, downregulated the hepatic mRNA expression of *Hmgcr* in both male and female offspring and decreased the hepatic mRNA expression of *Cpt1a* and *Acox1* in females. The postnatal diet significantly altered the circulating fatty acid concentrations, with sex-specific differences in genes that control lipid metabolism in the adult offspring following exposure to a high-LA diet in-utero [70]. In mice as well, the effect of a maternal high-fat diet on the metabolic phenotype was more obvious in male offspring, since increased body weight, impaired glucose tolerance, increased adipose tissue mass and hypertrophy, as well as a decreased circulating asprosin level (a fasting-induced adipokine), were only observed in males [71].

## 7. Effects of a Maternal Diet Enriched with Cholesterol and Fat on the Post-Natal Phenotype of F1 Offspring

### 7.1. Cardiometabolic Phenotype

As the maternal diet, specifically LA intake, is known to have significant repercussions on fetal development, it might lead to long-term consequences in the offspring, including the possibility of future metabolic and mental diseases [28], the cardiometabolic phenotype was studied. At birth, the survival rate was not statistically different between the H and C groups. Nevertheless, the H offspring had a reduced birthweight despite a significantly longer gestational period [35].

Litters were cross-fostered immediately after birth and equilibrated in terms of the number of pups per litter to obtain four groups of offspring—C-C, C-H, H-C and H-H—to account for maternal diet during lactation (the first letter corresponds to the maternal diet during the preconception and gestation periods and the second to the maternal diet during lactation). Due to the low number of females, it was not possible to consider the individuals’ sex-specific effects in this experiment. The H offspring had rapid catch-up growth so that the body and organ weights were no longer significantly different at weaning between the individuals in the four groups (Figure 1). In addition, no atherosclerotic lesions were observed in the aorta at this stage, despite lipid depositions being observed at birth [35]. The plasma glucose and HDL concentrations did not differ between groups, but the total cholesterol was significantly higher in the C-H offspring than in the H-H group. In addition, the C-H and H-H groups had significantly higher concentrations of total cholesterol than the C-C and H-C groups, respectively. Finally, the plasma triglyceride concentrations were significantly higher in the H-H groups versus the H-C and C-C groups [35]. Altogether, these data indicate an effect of the maternal diet during lactation and also an effect of the ante-natal period on the offspring phenotype. From weaning (5 weeks), all the animals were fed the control diet until 6 months of age to investigate whether a C diet is capable of reversing the phenotype. From the age of 3 months, the weights of the H-H and H-C rabbits were not significantly different, but these two groups became significantly heavier than the C-C and C-H groups. Over the first 6 months, the H-H and H-C offspring had a significantly higher food intake than the C-C and C-H pups. C-H individuals were also significantly heavier than C-C individuals (Figure 1 and Table 1). All these data show that it was not possible to reverse the phenotype of the offspring with a C diet.

In rats, it has been shown that high maternal adiposity induced by a high-fat or fructose diet during pregnancy programs a susceptibility to diet-induced hepato-steatosis [72]. Another study in Wistar rats showed that maternal exposure to the cafeteria diet during gestation promoted adiposity in offspring, which could be counteracted by taurine supplementation that reduced lipid deposition in both male and female offspring and altered patterns of hepatic gene expression, thereby reducing the detrimental effects of the maternal cafeteria diet [73]. Male (but not female) offspring from mothers fed the high-caloric obesity-inducing diet with casein protein demonstrated hyperphagia, obesity, dyslipidemia and hepatic triglyceride accumulation as adults compared with their controls in Sprague Dawley rats [74]. Offspring born from high-fat-fed mini-pig mothers were characterized, at one week, by hyperglycemia, hyperinsulinemia, severe insulin resistance and high liver and thymus glucose uptake, associated with thymocyte size and density, elevated weight gain, liver insulin resistance and steatosis in the first 6 months of life [75].

In rats, gestational exposure to maternal hyperlipidemia predisposed offspring to blood pressure elevation and sustained increases in leptin levels without significant changes in body weight [76]. Thus, to investigate blood pressure, all offspring were exposed to the H diet for 3 weeks at 176 days (6 months) of age. Food intake of the H diet did not differ between groups. At the end of the challenge, the systolic, diastolic and mean blood pressure were significantly increased in the C-C, C-H and H-C offspring. In contrast, only the mean arterial pressure was increased in the H-H animals [35] (i.e., high pressure in arteries, suggesting a loss of vascular elasticity).

Post-mortem examinations showed no statistical differences in organ weights between the four groups. But when analysis considered only the diet of F0 does, the liver and kidney relative weights to the whole-body weight were lower in the offspring from H dams (H-C and H-H) compared with the offspring from C dams (C-C and C-H). The fat mass was significantly higher in the offspring from H dams versus offspring from C dams. No atherosclerotic lesions were observed.

The cardiometabolic phenotype observed in the F1 offspring indicate an effect of the maternal diet during lactation and also an effect of the ante-natal period on the offspring phenotype, as fetuses from the H group exhibited dyslipidemia. Moreover, feeding with a C diet from the weaning until 6 months of age was not able to rescue the phenotype, as the H-H and H-C offspring had a significantly higher feed intake than the C-C and C-H pups, and the C-H individuals were also significantly heavier than the C-C individuals. The nutritional challenge of 3 weeks allowed revealing that the H-H individuals were characterized by a higher arterial pressure. It appears that the nutritional intervention after weaning is not sufficient to improve the cardiometabolic phenotype. These data corroborate those from publications by Gluckman et al [16,77], indicating that early intervention during the plasticity period is more efficient for improving the offspring phenotype.

### 7.2. Gonadal Phenotype

For the analysis of the F1 female gonadal phenotype in adulthood, the F1 females were exposed to the same diet throughout their in-utero lives until their weaning (i.e., H or C diet). At weaning, the F1 offspring were placed on either the C or H diet, leading to four new groups: C-Caw, C-Haw, H-Caw and H-Haw (with “aw” meaning the diet after weaning). Between 18 and 22 weeks of age, the females were mated with control males and then euthanized at day 28 of gestation.

The fertility was calculated as the ratio between the number of pregnant and non-pregnant female rabbits, whereas prolificacy was defined as the number of offspring per doe that gave birth. There was no significant difference in fertility and prolificacy between the C-Caw, C-Haw, H-Caw and H-Haw groups in the F1 females. Their ovaries were collected at the end of gestation [38]. The width of the ovaries was significantly reduced in does born to the H-Haw dams compared with the H-Caw group. In addition, the ovarian longitudinal surface area was significantly lower in the F1 H-Haw does than in the F1 C-Caw does. Thus, in humans, a young woman with small ovaries is more likely to have difficulty carrying a normal pregnancy to term because she will produce fewer eggs [78]. The number of primordial, primary and secondary follicles was not significantly different between groups. Nevertheless, a significantly higher number of atretic follicles was observed in the H-Caw, C-Haw and H-Haw ovaries compared with the C-Caw group, indicating a deleterious effect from both prenatal and postnatal maternal diets. Interestingly, the number of atretic follicles was not significantly different between the H-Haw, H-Caw and C-Haw groups, demonstrating similar effects from the prenatal and postnatal diets. There was also a positive correlation between the follicular atresia and plasma total cholesterol or triglyceride concentration and a negative correlation between the follicular atresia and plasma testosterone (Table 1). Despite those observations, the expression of candidate genes involved in ovarian development, inflammation and oxidative stress was not different between groups. Adult F1 offspring fed the H diet and the ones born to dams fed the H diet had a significantly decreased ovarian area and increased follicular atresia, suggesting that ovarian function is sensitive to prenatal and postnatal H diets without affecting fertility. These data are consistent with previous studies related to the administration of a high-fat diet. Indeed, a high-fat diet was shown to reduce the ovarian surface area and to increase ovarian atresia in mice, rats and rabbits, impairing follicular reserve [36,79,80]. The higher number of atretic follicles suggests a possible increase in apoptotic mechanisms during folliculogenesis. Unfortunately, this hypothesis has not been investigated using TUNEL in F1 adults. As for the F0 H does, the explanations for the increase in atretic follicles might be related to the estradiol concentrations, known for its anti-apoptotic role [51], although the hormonal profile has not been investigated at this stage, or to circulating blood cholesterol, since follicular atresia has been observed as well in rats fed a high-cholesterol diet [52]. Moreover, a depletion of antioxidant enzymes associated with an increase in lipid peroxidation and apoptotic markers has been described in this model but with alterations in female serum hormones [52]. The molecular analysis of the ovarian expression of genes involved in ovarian development and oxidative stress was not significantly different between the groups. Nevertheless, the positive correlation between cholesterol and triglycerides and the number of apoptotic follicles was consistent with previous data highlighting that cholesterol abundance promotes reactive oxygen species and exacerbates apoptotic cell death in the ovarian cell [81]. More recent studies have shown that a high-fat maternal diet affects the expression of genes related to follicle growth in offspring, such as *AAT *(*α1-antitrypsin*)**, *AFP *(*α-fetoprotein*)** and *GDF-9 *(*growth differentiation-9*)**, reducing the number of follicles and altering the development of follicles [80]. Additionally, a high-fat maternal diet also affects ovarian health by inducing ovarian oxidative stress and cellular apoptosis, which collectively can impair the reproductive potential of female offspring.

For the F1 male gonadal phenotype, F1 males were exposed to the H or C diet throughout their in-utero lives until weaning, leading to C-C and H-H groups. Then, the males received a C diet until 37 weeks of age. At this stage, only the testes and semen from the C-C and H-H males were analyzed. The H-H F1 male offspring had significantly lighter testes and epididymis compared with the C-C males without differences in seminiferous epithelium thickness between groups. No significant differences were observed in sperm concentration, sperm DNA integrity or sperm cholesterol and phospholipid membrane composition. The free testosterone plasma concentrations, however, were reduced in the HH vs. C-C males [37] (Figure 1 and Table 1). Male gonadal development starts in-utero, and thus nutritional stress during this period may challenge testicular and epididymis development. As the testes develop slowly from birth to 5 weeks of age in a rabbit, the maternal H diet throughout gestation and the lactation period could affect gonadal development, as shown in obese rabbits [82]. Maternal overnutrition in sheep has also been associated with alterations in the reproductive functions of male offspring, such as lower plasma testosterone concentrations and a reduced testicular volume [83]. Therefore, the maternal dyslipidemia could challenge the concentration of testosterone required for Sertoli cell proliferation and testis development [84]. Moreover, a decrease in testosterone production by Leydig cells could contribute to the low testosterone concentration, as observed in obese men [85]. In addition to hormone effects, other mechanisms such as oxidative stress and epigenetic changes could be involved in fetal programming and also impair reproductive functions in offspring [86,87,88]. However, no effect was observed on the sperm concentration, sperm DNA integrity or sperm membrane in our study.

Altogether, these data underline the importance of maternal nutrition in offspring development and gonadal function. It should be noted, however, that rabbit gonadal development differs in terms of timing and regulation from that of humans and rodents. In rabbit does, meiosis occurs postnatally in the first 15 days after birth in the ovary, and folliculogenesis starts afterward, whereas these events occur during gestation in humans [89,90]. Thus, effects observed on follicular atresia in the H-Haw group might not be due to direct effects on follicles but rather to effects on the ovaries prior to meiosis. In male rabbits, male testicular differentiation that occurs at mid-gestation is also prolonged postnatally, but since only the C-C and H-H groups were compared, is it not possible to differentiate gestational from neonatal effects in the present studies.

## 8. Identification of the Exposure Windows Responsible for the Fetal-Placental Phenotype: What Is the Importance of the Preconception and Gestation Windows?

We have previously shown that fetuses from dams fed before and throughout gestation with a H diet were characterized by intrauterine growth retardation associated with placental abnormalities and lipid metabolism disorders [34]. Maternal exposure to the H diet occurred from puberty until the end of gestation, and the effects of preconception exposure were not discriminated from those of gestational exposure only. To evaluate the effects of each window of exposure on feto-placental development, we targeted the following two compartments, placenta (trophoblast) of fetal origin and decidua of maternal origin, to study the ability of the fetus to modify its placenta in response to the maternal environment and the ability of the maternal decidua to adapt to the embryonic origin of the fetus.

Embryos 1 d.p.c. from C or H donors were surgically transferred to the uteri of synchronized C or H recipients to continue their gestational development until 28 d.p.c. (Figure 3). Each recipient received, in each of the uterine horns, embryos from different origins (5–6 embryos/horn). These transfers resulted in the following four combinations: C/C, C/H, H/C and H/H, with the first letter corresponding to the embryo donor (d) and the second to the recipient (r) (Figure 3).

### 8.1. Feto-Placental Phenotype

As described in our earlier studies without embryo transfer, fetal hypotrophy was observed in the H/H group compared with the C/C group [39], which enabled us to exclude a potential effect of embryo transfer on this parameter. As expected, placental efficiency was reduced in the H/H versus C/C groups. The H/H liver and kidneys were also lighter than those of the controls. Such hepatic hypotrophy was shown in male offspring at weaning from C56BL/6N mice fed a high-fat diet inducing maternal obesity through possible premature hepatic aging [91] as well as in weaned offspring rats, but the liver weight increased in adult males [66]. The intermediate groups, H/C and C/H, were used to explore the critical window of exposure. The H/C fetuses had a normal development. The C/H fetuses’ weights were also similar to those of the controls, but their lengths were significantly increased. Placental efficiency was decreased, with a heavier labyrinthine area, in the C/H fetuses (Figure 4).

### 8.2. Quantification and Profiles of Placental FA

Placental FA concentrations including phospholipids and neutral lipids were quantified at 28 d.p.c. [39]. The FA concentrations of placental phospholipids were not significantly different between the four groups. On the other hand, the FA concentrations of neutral (or storage) lipids were higher in the rH recipient groups (C/H and H/H) than in the rC recipient groups (C/C and H/C) (Figure 4).

The principal component analysis (PCA) of lipid profiling from the placental labyrinthine zone showed that the recipient dam’s diet had a very strong impact on placental tissues, as the C/H and H/H placentas had higher concentrations of FA and were qualitatively richer in polyunsaturated FA (PUFA, 18:2ω6 and 20:3ω6) and poorer in monounsaturated FA (MUFA) than the C/C and H/C placentas. The PCA of the H/C FA profiles were different from those of the C/C group, with a deficit in PUFA ω-6 (18:2ω6) and an increase in SFA. The C/H placentas had a more disturbed FA profiling than those of the H/H group, with a higher MUFA content.

The FA profiles of the H/C and C/C deciduae were not significantly different, suggesting that the embryonic origin has little importance if the gestational diet is well balanced. In contrast, the FA profiles of the C/H decidua were closer to the C/C group than the H/H group, suggesting that the decidua adapts to the fetal origin when it is healthy. In contrast, the membranes of the H/H deciduae were richer in SFA and less so in PUFA.

The multiple factor analysis (MFA) combining different datasets, namely feto-placental biometry, placental gene expression and placental FA concentrations and profiles (composition of FA expressed as a percent of total FA), showed that although maternal diet during gestation played a prominent role, the four groups were quite distinct [39]. Thus, the H/C group was completely dissociated from the C/C group, and the C/H group was distinct from the H/H group. The feto-placental units carried by C recipients (rC) were characterized by placental profiling (PL and LN) in FA enriched in MUFA and SFA, a stronger gene expression of placental nutrient transporters, while the neutral lipid and phospholipid FA concentrations and a higher PUFA/SFA ratio defined the effect of the H (rH) recipients. Finally, the origin of the embryo was correlated with biometric data positively for feto-placental units from C donors (dC) with higher fetal, liver and kidney weights and with an increase in the FA concentration of PL, while the origin dH were characterized by a higher MUFA content.

MFA combining fetal placental biometric data, liver gene expression and liver FA concentrations and profiles showed, as for the placenta, that although the maternal diet during gestation played a key role, the four groups were quite distinct. Therefore, the origin of the embryo was also important for the offspring hepatic phenotype in late gestation. Higher gene expression (*IRS1*, *SREBP2*, *SCD5*, *PCTP*, *IR* and *PPAR-α*), as well as higher long-chain DHA and MUFA contents (C18, C20 and C24) of phospholipids and hepatic neutral lipids, contributed to groups in the recipient dams (rC), while higher levels of linoleic acid, PUFA-ω6 and alpha-linolenic acid of phospholipids and neutral lipids and higher PUFA/SFA and PUFA ω6/ω3 ratios characterized the groups in the H recipient dams.

The origin of the H embryo (dH) was correlated with a higher contribution from MUFA, FA concentrations of neutral lipids and phospholipids, and indices of desaturation delta6 and delta9, as shown in rodent studies where high maternal adiposity was induced by a high-fat or fructose diet during pregnancy programs hepato-steatosis in offspring, through a significant effect on FA elongation and delta9 desaturase activity [72], while normal liver and kidney biometrics, as well as higher levels of SFA and PUFA, contributed to groups from dC donors (Figure 4) [39].

Thus, the embryos collected from the dH evolved favorably at first when transferred to a control recipient (H/C group), with normal development at term despite FA placental profiling remaining disturbed.

These results show that maternal exposure to an H diet during these two periods leads to distinct and sometimes subtle effects that could nevertheless impact the health of offspring in adulthood. Although the H/C and C/C individuals were not similar, the differences that persisted between these two groups seemed negligible. In contrast, the C/H and H/H groups had more severe phenotypic impairments.

## 9. Conclusions

The objective of this work was to summarize 15 years of research about the effects of a high-fat high-cholesterol maternal diet, mimicking the overconsumption of lipids in European women, on the maternal phenotype and offspring at the embryonic, neonatal, weaning and adult stages, considering sex-specific effects while trying to discriminate the window or windows of vulnerability related to this exposure in a rabbit model.

We showed that a diet high in fats (high linoleic acid) and cholesterol induced dyslipidemia and increased fat mass but without inducing overweight in rabbit. Before mating, these fatty F0 H females also exhibited aortic atherosclerotic lesions and altered follicular growth and hormonal responses. In the offspring, this diet disrupted the embryonic phenotype and induced fetal hypotrophy associated with sex-specific disturbances of the fetal-placental unit. In adulthood, the offspring from H dams were heavier, being hyperphagic with increased blood pressure, and male and female gonadal development was slightly disturbed.

The identification of vulnerability windows via embryo transfers shows that the gestational maternal diet plays a predominant role in the feto-placental phenotype and that there is undeniably preconception programming. These two periods therefore seem to be candidates for intervention in the context of obesity or overweight to limit the adverse consequences on the fetal placental unit. Several intervention strategies have already been explored in the literature through the use of animal models or through clinical trials.

These strategies range from preconceptional weight loss [27] to nutritional supplementation [92] and exercise. More invasive approaches such as bariatric surgery may be proposed but are restricted for cases of severe or morbid obesity.

One of the first strategies of intervention is to promote healthy habits through lifestyle changes, which combine dietary interventions and physical exercise during these periods and show maternal-fetal benefits [93]. In fact, a maternal high-fat diet (60% kcal from fat) was shown to induce dysregulation of offspring liver glucose metabolism in mice through a mechanism involving increased reactive oxygen species, carbonylation and inactivation of histone H3 lysine 4 (H3K4) methyltransferase, leading to decreased H3K4me3 at the promoters of glucose metabolic genes, and all were counteracted when the high-fat diet-fed dams exercised during pregnancy [94]. However, dietary interventions, such as limiting dietary fat intake (<35%) with appropriate fatty acid intake during the gestational period, appear to be one effective type of intervention to improve the maternal metabolic environment during pregnancy [13]. Thus, in mice, a maternal, balanced low-fat diet fully compensated for the detrimental effects of a maternal high-fat diet on glucose metabolism, insulin tolerance, circulating insulin, dyslipidemia and body weight gain in male offspring by changing the gene expression profile involved in the peroxisome proliferator-activated receptor signaling pathway [95].

The weight loss tested can correct the metabolic health of male offspring but induce negative effects on the olfactory performance of these individuals [27]. The benefit/risk ratio of weight loss before conception may therefore be uncertain for offspring, especially if nutritional deficiencies appear [96] or if a rebound in weight gain is observed after conception [97].

To our knowledge, nutritional supplementation such as long-chain ω-3 PUFA intake during the preconception period was not explored in the context of being overweight or obese. As for the supplementation of fish oil, which is rich in ω-3 PUFA, in overweight or obese mothers during pregnancy, it does not show a positive effect on the body composition of the infant and remains to be determined whether this is confirmed or not in the longer term [98]. Nevertheless, in mice, leucine supplementation in high-fat diet-fed dams resulted in an anti-obesity phenotype accompanied by improved glucose tolerance in male offspring challenged with postnatal high-fat feeding, probably through the activation of signaling involving fibroblast growth factor 21 (FGF21), a hepatokine associated with glucose homeostasis, in the adipose tissue of offspring [99]. The isocaloric exchange of casein protein for yellow pea protein (YPPN) in a high-calorie obesity-inducing diet did not protect against obesity but did improve several aspects of lipid metabolism in adult male offspring, including the serum total cholesterol, LDL and VLDL cholesterol, triglyceride (TG) and hepatic TG concentrations, suggesting that the exchange of proteins in a deleterious maternal diet selectively protects male offspring from the malprogramming of lipid metabolism in adulthood [74].

These different interventional trials are interesting avenues, but they also show the difficulties in assessing the benefit and risk of these strategies for the health of the offspring, which implies both prenatal and long-term phenotyping to detect the occurrence or absence of noncommunicable diseases. Owing to the length of the intergenerational period in humans, it is essential to be able to answer this question through appropriate animal models. Finally, multidisciplinary medical management, including nutritional assessment, is essential to the planning of a pregnancy to improve both maternal health and that of their progeny.

## Figures and Tables

**Figure 1 ijms-24-14547-f001:**
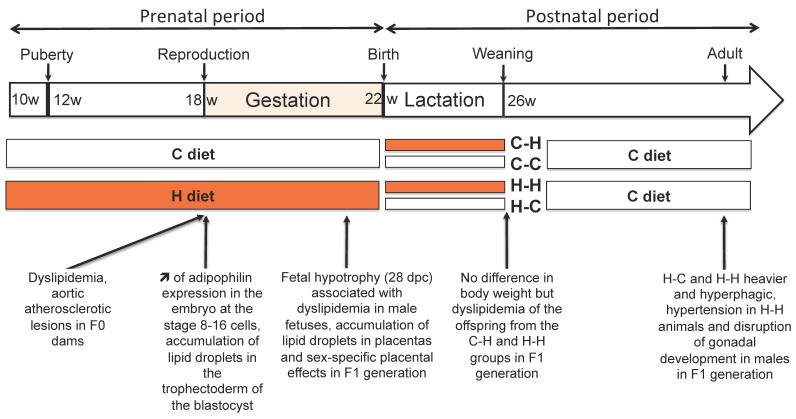
**Experimental protocol of exposure to the maternal diet enriched with cholesterol and fat and the main results.** At the age of 10 weeks and throughout gestation, females were fed ad libitum with a high fat and cholesterol (H) diet or a control diet (C). At birth, the litters were cross-fostered and equilibrated at six pups per litter to obtain four groups: C-C, C-H, H-C and H-H to take into account the maternal diet during lactation, the first letter corresponds to the maternal diet during the preconception and gestation periods and the second to the maternal diet during lactation. Then, the F1 offspring received a control diet after weaning. The main results are specified according to the stage compared to the control diet (C or C-C).

**Figure 2 ijms-24-14547-f002:**
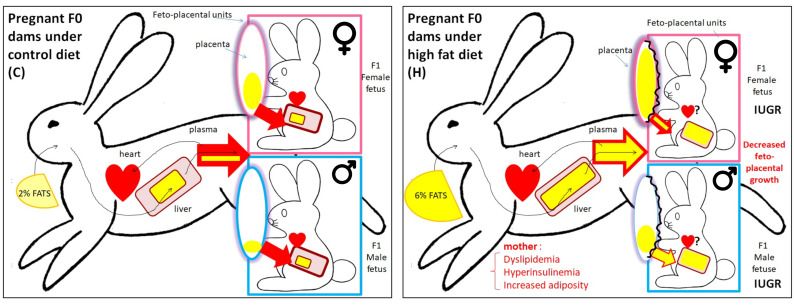
**Schematic representation of the effect of a diet high in cholesterol and fats on the transfer of fatty acids from the mother F0 to the placenta and then to the male or female F1 fetuses in the rabbit model, at the prenatal stage.** On the left side, the maternal-fetal flows when the doe is fed the control diet (C), and on the right side, the maternal-fetal flows when the doe is fed the fatty diet (H). Female fetus is in a pink frame and male fetus is in a blue frame. The lipids and their accumulation are represented by yellow zones, all the more voluminous as the storage or the flows are important. IUGR: intrauterine growth restriction.

**Figure 3 ijms-24-14547-f003:**
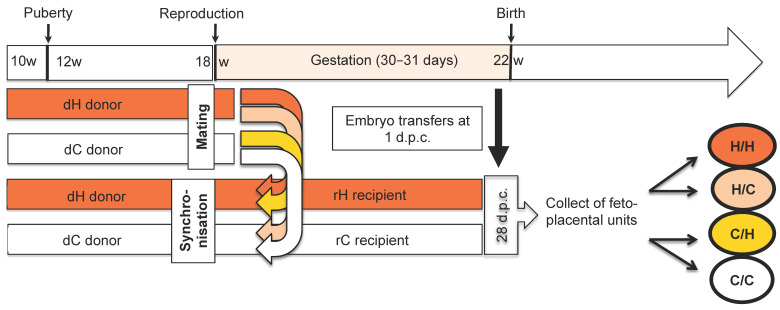
**Experimental protocol of embryo transfer at 1 day post-coitum.** At the age of 10 weeks, females were fed ad libitum with a diet enriched in fats and cholesterol (H) or a control diet (C). Several females were superovulated and mated with a control male, and called embryo donors (d), while the other females were synchronized by hormonal treatment, and called embryo recipients (r). To evaluate maternal effects, five to six embryos from H donors (dH) were transferred to the right horn and five to six embryos from C donors (dC) were transferred to the left horn of each C recipient (rC) and H (rH), to give the following 4 groups, H/C, H/H, C/C and C/H, respectively. Then, gestation continued in the uterus of the recipient females until the 28th day post-coitum (d.p.c.) when the fetal-placental units H/C, H/H, C/C and C/H were collected.

**Figure 4 ijms-24-14547-f004:**
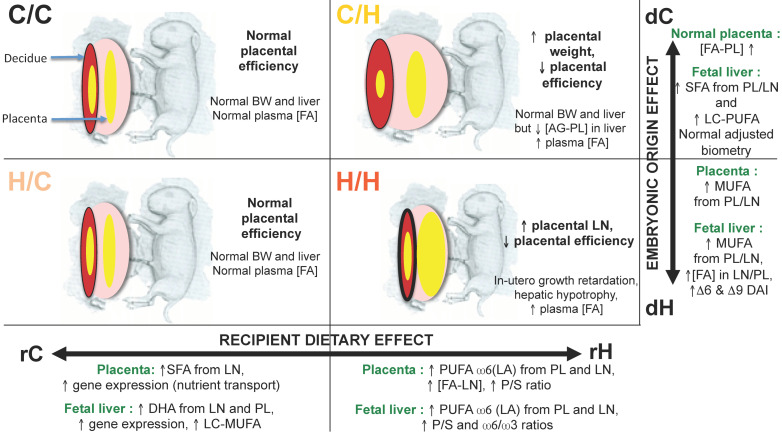
**Summary of feto-placental effects of maternal exposure to a H or C diet during preconceptional and/or gestational periods.** C/C: embryo from C donors transferred in C recipient females; C/H: embryo from C donors transferred in H recipient females; H/C: embryo from H donors transferred in H recipient females; H/H: embryo from H donors transferred in H female recipients. BW: body weight; DAI: desaturase index; DHA: docoxahexaenoic acid; FA: fatty acids; LC: long chain; LN: neutral lipids; MUFA: monounsaturated FA; PL: phospholipids; P/S: PUFA/SFA ratio; PUFA: polyunsaturated FA; SFA: saturated FA. In the placenta and the decidua, the lipids and their accumulation are represented by yellow zones (neutral lipids, NL) or the thickness of the lines (phospholipids as membranes, PL), all the more voluminous or thick as the storage is important. Adapted from Rousseau-Ralliard et al., 2021 [37].

**Table 1 ijms-24-14547-t001:** Sex-specific effects of high-cholesterol and high-fat diet during the prenatal and postnatal periods.

	F1 Male Offspring	F1 Female Offspring
**Prenatal period** F0 does were exposed to H or C diet from 10 weeks of age until 28 d.p.c. to obtain F1 feto-placental unit	**Blood biochemical parameters (n = 12):** 🡭 total cholesterol in H vs. C (*p* < 0.05) 🡭 of FA in H vs. C (*p* < 0.01) 🡭 MUFA in H vs. C (*p* < 0.05) 🡭 PUFA in H vs. C (*p* < 0.001)	**Blood biochemical parameters (n =11):** - - - 🡭 PUFA in H vs. C (*p* < 0.01)
	**Hepatic FA profiles (n = 6):** 🡭 of FA storage in H vs. C (*p* < 0.001) 🡭 (*p* < 0.05) PUFA (*p* < 0.001) storage in H vs. C	**Hepatic FA profiles (n = 6):** 🡭 of FA storage in H vs. C (*p* < 0.01) 🡭 (*p* < 0.05) PUFA (*p* < 0.001) storage in H vs. C
	**Placental lipid profiles (n = 5):** 🡭 of total cholesteryl esters in H vs. C (*p* < 0.01) 🡭 of triacylglycerol in H vs. C (*p* < 0.05) **FA placental membranes:** - - - **FA placental storage:** 🡭 of total FA in H vs. C (*p* < 0.01) - - 🡭 of total PUFA in H vs. C (*p* < 0.01)	**Placental lipid profiles (n = 5):** 🡭 of total cholesteryl esters in H vs. C (*p* < 0.01) - **FA placental membranes:** 🡭 of FA (*p* < 0.01), SFA (*p* < 0.01), MUFA (*p* < 0.01) and PUFA (*p* < 0.01) in H females compared to H males 🡭 of PUFA in H vs. C (*p* < 0.01) **FA placental storage:** 🡭 of total FA in H vs. C (*p* < 0.05) 🡭 of total FA (*p* < 0.05), SFA (*p* < 0.05), MUFA (*p* < 0.05) and PUFA (*p* < 0.01) in H females compared to H males 🡭 of total PUFA in H vs. C (*p* < 0.01)
**Postnatal period**	**Gonadal phenotype at 37 weeks of age** **(n = 8H-H and 7C-C) *** Lighter testes and epididymes in H-H vs. C-C (*p* = 0.03 and 0.015, respectively) No significant differences in sperm concentration, sperm DNA integrity and sperm membrane composition 🡶 of plasma-free testosterone concentrations in H-H vs. C-C (*p* = 0.05)	**Gonadal phenotype at adulthood** **(n = 7C-Caw, 14H-Caw, 9H-Caw, 8H-Haw) #** 🡶 of width of the ovaries and ovarian longitudinal surface in H-Haw vs. H-Caw group (*p* < 0.05) 🡶 of ovarian longitudinal surface in H-Haw vs. C-Caw group (*p* = 0.03) No significant differences in the number of primordial, primary and secondary follicles between groups. 🡭 of atretic follicle number in H-Caw, C-Haw (*p* < 0.001) and H-Haw (*p* < 0.01) ovaries compared to the C-Caw group No significant differences of the number of atretic follicles between H-Haw, H-Caw and C-Haw groups Positive correlation between follicular atresia and plasma total cholesterol (*p* = 0.03) or triglyceride concentration (*p* = 0.01) Negative correlation between follicular atresia and plasma testosterone (*p* = 0.02)
**Postnatal period**	**Cardiometabolic phenotype (n = 26) $** At 1 month of age (weaning period), 🡭 of total cholesterol concentration in C-H vs. H-H group (*p* < 0.05) 🡭 of total cholesterol concentration in C-H and H-H vs. C-C and H-C groups (*p* < 0.05) 🡭 plasma triglyceride concentrations in H-H vs. H-C and C-C groups (*p* < 0.05) At 3 months of age, of weight 🡭 of the H-H and H-C vs. C-C and C-H groups (*p* < 0.0001) 🡭 of weight in C-H g vs. C-C offspring (*p* < 0.05) At 6 months of age, 🡭 H-H and H-C offspring vs. C-C and C-H groups (*p* < 0.05) At 6 months of age, all offspring were exposed to the H diet for 3 weeks. At the end of the challenge, 🡭—systolic, diastolic, and mean blood pressure in C-C, C-H, and H-C offspring (*p* < 0.01) 🡭 mean arterial pressure in H-H animals (*p* < 0.05)	

* F1 females were exposed to the same diet throughout their in-utero life until weaning, leading to C-C and H-H groups. Then, the males received a C diet. # F1 females were exposed to H or C diet during their in utero lives until weaning. At weaning, F1 offspring were placed on either C or H diet, leading to four groups: C-Caw, C-Haw, H-Caw and H-Haw (aw = after weaning). Females were bred with control males between 18 and 22 weeks of age and then euthanized at day 28 of gestation to collect ovaries. $ F1 generation was exposed to H or C diet during their in utero lives. At birth, litters were cross-fostered and equilibrated to consider what the diet of F0 does during lactation, leading to four groups: C-C, C-H, H-C and H-H. After weaning, F1 generation received a control diet. Due to the low number of females, it was not possible to take into account the sex effects at adulthood. C: control diet; d.p.c.: days post-coitum; FA: fatty acids; H: high-cholesterol and high-fat diet; MUFA: monounsaturated fatty acids; PUFA: polyunsaturated fatty acids; SFA: saturated fatty acids.

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
