# Peer review of "The Effect of Maternal Exposure to a Diet High in Fats and Cholesterol on the Placental Function and Phenotype of the Offspring in a Rabbit Model: A Summary Review of About 15 Years of Research"

_ijms, 2023, doi:10.3390/ijms241914547_

Round 1

Reviewer 1 Report

1)     Insert the use of your Rabbit model explaining why you chose this model respect to other animal models, in particular respect to mice and rats that have a shorter gestational time.

2)     Sex differences in dislypidemia are summarized in Figure 1 and 2; however, it is difficult to understand effects on males and females undergoing different experimental protocols (H-H, H-C, C-C and C-H). Can you provide a table with numbers and statistics to explain better this issue, subdividing prenatal and postnatal effects of different diets/combination of diets on males and females?

3)     Rabbits born from H- dams or with H-diet during lactation can recover from their problems after a long-term C-diet? Have you tried to do this experiment or compensative experiments to revert this phenotype?

Author Response

Thank you very much for taking the time to review this manuscript. Please find the detailed responses below and the corresponding revisions indicated in red with highlighted in blue in the case of a discussion in the re-submitted files

  • Insert the use of your Rabbit model explaining why you chose this model respect to other animal models, in particular respect to mice and rats that have a shorter gestational time.

We dedicated the part 2 to the rabbit model and make comparisons with the human and murine species. To better justify our choice several sentences has been added in red, line 108, 111, 121, 140 p2 and line 151 p3, as well as additional references. Moreover, a sentence of conclusion has been added line 153 at the end of this 2nd part.

  • Sex differences in dyslipidemia are summarized in Figure 1 and 2; however, it is difficult to understand effects on males and females undergoing different experimental protocols (H-H, H-C, C-C and C-H). Can you provide a table with numbers and statistics to explain better this issue, subdividing prenatal and postnatal effects of different diets/combination of diets on males and females?

To clarify the different protocols, two arrows have been added on Figure 1 to separate the prenatal period from the postnatal period. Moreover, F0 and F1 generation has been specified.

On Figure 1, “and females” in the square has been supressed from « H-C and H-H heavier and hyperphagic, hypertension in H-H animals and disruption of gonadal development in males and females in F1 generation » since the protocol of diet exposure was not exactly the same.

Moreover, always to clarify the different protocols, the legend of Figure 2 « Figure 2: Schematic representation of the effect of a high fat diet on the transfer of fatty acids from the mother to the placenta and then to the male or female fetuses in the rabbit model”, has been completed by « Figure 2: Schematic representation of the effect of a diet high in cholesterol and fats on the transfer of fatty acids from the pregnant F0 dams to the placenta and then to the male or female F1 fetuses in the rabbit model, at the prenatal stage”.

Please, note that male and female F1 fetuses studied at the prenatal period are obviously different from males and females follow-up at adulthood.

The authors thank the reviewer for his suggestion to explain better sex differences at fetal stage and postnatal period, for this purpose, a table has been added (Table 1 p10), including numbers and statistics as requested. Table 1 is indicated in the manuscript in red, line 267 p6, line 306 p7, line 438 p 11, line 505 p 12 and line 541 p 13. Moreover, to clarify the protocol of exposure, a sentence was added at the beginning of the paragraph concerning female results on gonadal phenotype line 438 p12. “For the analysis of the F1 female gonadal phenotype in adulthood, F1 females were exposed to the same diet throughout their in utero life until their weaning, i.e. H or C diet. At weaning, F1 offspring were placed on either C or H diet, leading to 4 new groups C-Caw, C-Haw, H-Caw and H-Haw groups (with “aw” meaning diet after weaning). At 18 to 22 weeks of age, females were mated with control males and then euthanized at day 28 of gestation ”.

Please note that the post-weaning development including blood pressure, weight, food intake, blood biochemistry parameters was monitored on 26 offspring but due to the low number of females, it was not possible to take into account the individuals’ sex-specific effects (Picone et al, 2010; DOI10.1016/j.theriogenology.2010.08.015). A sentence has been added in the paragraph 7.1, 421  p 11.

3)     Rabbits born from H- dams or with H-diet during lactation can recover from their problems after a long-term C-diet? Have you tried to do this experiment or compensative experiments to revert this phenotype?

In fact, rabbits born from H dams or indirectly fed with H diet during lactation, then received a C diet from weaning until 6 months of age (see paragraph 7.1). These groups correspond to H-C and C-H, i.e. the first letter indicates the maternal diet during the preconception and gestation periods and the second, the maternal diet during lactation.

The sentence in the paragraph 7.1 “At weaning (5 weeks), all the animals were fed the control diet until puberty » has been suppressed. Another one has been added line 432 p11 “From weaning (5 weeks), all the animals were fed the control diet until 6 months of age to investigate whether a C diet is capable of reversing the phenotype».

These results are summarized in the paragraph 7.1. “From the age of 3 months, the weight of the H-H and H-C rabbits was not significantly different but these two groups became significantly heavier than the C-C and C-H groups. Over the first 6 months, H-H and H-C offspring had a significantly higher food intake than C-C and C-H pups. C-H individuals were also significantly heavier than C-C (Figure 1) (Table 1). “

A sentence of conclusion has been added “All these data show that it was not possible to reverse the phenotype of the offspring with a C diet”. Line 439 p11.

Reviewer 2 Report

This review focuses on the authors' rabbit model of a maternal high-fat diet with analysis of the effects on the mother, embryo, fetus, placenta and postnatal offspring.  Overall, it gives a good summary of their specific findings, but little about the potential mechanisms responsible or how this model compares with other studies that have investigated the effects of maternal high-fat feeding.

1. The title could be shorted by deleting the second half after the colon.

2. In the Abstract, line 10, state 'in part due to overconsumption of lipids'.

3. Check the position and numbering of the figures.  There are two Figure 2 and no Figure 4 (line 379).  Define 's' in Figure 1 (week rather than semaine).

4. Line 154-155, please discuss why was there no change in weight gain between the mothers on the control and high fat diets.

5. Lines 180 and 192, clarify the changes in adipophilin in response to the H diet as these sentences suggest decreased and increased expression in the developing embryo.

6. Line 202, indicate the gestational length of the rabbit so that 28 dpc can be understood in context.

7. Lines 202-203, were there any changes in placental weight between the groups?  Or changes in placental micro-structure (vascularity, surface area, effective diffusion distance, placental cell types)?

8. Was there any evidence of fetal brain-sparing in this model of suboptimal maternal nutrition?  And any evidence about relative organ weights or adiposity in the pre- and postnatal offspring?

9. Lines 325-327, is it unlikely that fertility would be affected if no difference in follicle numbers was observed between the groups?  What mechanism might be responsible for the greater number of atretic follicles?  Has this been investigated in similar models?

10. Is anything known about the endocrine profiles in the mother and offspring of this model?  Please include if available and discuss as a potential underlying mechanism for the metabolic, cardiovascular and reproductive outcomes.

11. Wherever relevant, include more discussion of the findings (rather than detailed results, more about what they mean for the mother, placenta and offspring overall), potential mechanisms responsible, and how the findings compare to those from similar models of maternal high-fat feeding.

There are just a few grammatical and spelling errors to correct.  For example, line 75 'temporary', line 171 'concentrations' given twice.

Line 86, could edit 'The aim of this review is to summarise ...'

Author Response

Thank you very much for taking the time to review this manuscript. Please find the detailed responses below and the corresponding revisions indicated in red with highlighted in blue in the case of a discussion in the re-submitted files

  1. The title could be shorted by deleting the second half after the colon.

Thanks to the reviewer for his suggestion. Nevertheless, the authors believe that it is important to keep « Summary review about 15 years of research » to guide the readers because it is not a manuscript with new data, but a compilation of data from several articles by the authors on this subject over a period of 15 years.

  1. In the Abstract, line 10, state 'in part due to overconsumption of lipids'.

« in part » has been added line 10 p1

  1. Check the position and numbering of the figures.  There are two Figure 2 and no Figure 4 (line 379).  Define 's' in Figure 1 (week rather than semaine).

There were indeed errors in the numbering of the figures, the 2nd figure 2 was renamed to figure 3 and the previously noted figure 3 was renamed to figure 4. They were also cited adequately in the text.

Figure 1 has been removed from Line 262 and 289 of the manuscript (submitted version), now line 312 p 8 of the revised manuscript.

Line 379 of the manuscript (submitted version), new line 596 p 15 of the revised manuscript  : the text corresponds to the identification of the exposure windows and consequently to the Figure 4

« s » is a mistake and has been replaced by w (weeks) on Figure 1 and 3

  1. Line 154-155, please discuss why was there no change in weight gain between the mothers on the control and high fat diets.

The maternal phenotype has been completed line 176 p4 by « but H does had a significantly higher body fat content compared to C group. »

This sentence has been added line 177 p 4 to discuss the results. « The absence of excess weight in fatty H females at 18 weeks of age could be explained by adaptations of food intake to their energy needs. In fact, food intake was reduced in H females compared to C females (data not shown) whereas the average weekly energy intake was higher in H compared to C does. »

  1. Lines 180 and 192, clarify the changes in adipophilin in response to the H diet as these sentences suggest decreased and increased expression in the developing embryo.

The sentences of line 180 and 192 p5 (now line 215 p5 and 227 p6) are in agreement with the results published by (Picone et al., 2011) and (Tarrade et al., 2013), i.e. at 16-20 cell stage, adipophilin gene expression was increased in embryos from H diet (Picone et al., 2011) whereas at blastocyst stage the adipophilin gene expression was reduced (ref Tarrade et al., 2013), although numerous lipid droplets were observed in their trophoblastic layer.

To discuss this point, this sentence, below, has been added at the end of the paragraph 5, line 232 p 6 and highlighted in blue.

“The maternal H diet affected the trophoblast at the blastocyst stage characterized by an increase of lipid contents and a deregulation of gene expression in the blastocyst, as observed in bovine embryo, sensitive to maternal hyperlipidemia [55]. The difference in Adipophilin gene expression observed between the 16-20 cell and the blastocyst stages could be due to a rapid regulatory loop aimed at limiting the accumulation of droplets in the embryo, as it is known that the decrease of adipophilin reduces the formation of lipid droplets [56]. Furthermore, it could also be an adaptive response to the maternal H environment related to the contents of uterine fluid, because the H environment differed from the C group [57] or to an alteration of follicular fluid contents in oocyte as evidenced in humans [58] and cattle [59]. »

  1. Line 202, indicate the gestational length of the rabbit so that 28 dpc can be understood in context.

Line 247 p6: to precise the gestational length, « gestation lasts 30-31 days in rabbits » has been added in brackets.

  1. Lines 202-203, were there any changes in placental weight between the groups?  Or changes in placental micro-structure (vascularity, surface area, effective diffusion distance, placental cell types)?

Lines 202-203 (submitted manuscript) now line 248 p6 of the revised manuscript correspond to placental data obtained by ultrasound so « measured by ultrasound » has been added.

The sentence line 254 p6 of the revised manuscript has been completed by « and placental weight was reduced in H group ». 

Concerning the « placental micro-structure », stereology was not performed in this particular study.

  1. Was there any evidence of fetal brain-sparing in this model of suboptimal maternal nutrition?  And any evidence about relative organ weights or adiposity in the pre- and postnatal offspring?

In these studies, the head length was not measured throughout the gestation nor the cerebral blood flow, nor the brain weight at 28 dpc, so we cannot conclude whether a fetal brain-sparing occurs in response to maternal H diet.

Concerning the relative organ weights, at 28 dpc, liver and heart weights were significantly reduced but their weight related to total fetal weight were unchanged. A sentence has been added line 256 p 6 « Liver and kidney relative weight to body weight were unchanged. »

At the end of paragraph 7.1 now p 11, two sentences have been added. “Post-mortem examinations showed no statistical differences in organ weights between the 4 groups. But when analysis considered only the diet of F0 does, liver and kidney relative weights to whole body weight were lower in offspring from H dams (H-C and H-H) compared to offspring from C dams (C-C and C-H). Fat mass was significantly higher in offspring from H dams vs offspring from C dams. No atherosclerotic lesions were observed. .”

  1. a- Lines 325-327, is it unlikely that fertility would be affected if no difference in follicle numbers was observed between the groups?  9. b- What mechanism might be responsible for the greater number of atretic follicles?  9.c- Has this been investigated in similar models?

9.a- We agree with the reviewer, as there was no difference in terms of follicle numbers, it is unlikely to observe an effect on fertility. In fact, in the study from Léveillé et al. 2014 (now reference 38), the fertility was calculated as the ratio between the number of pregnant on non-pregnant female rabbits. Prolificacy was defined as the number of offspring per doe that gave birth. There was no significant difference in fertility and prolificacy between C-Caw, C-Haw, H-Caw and H-Haw in the F1 females (with "aw" meaning diet after weaning). This point has been added line 488 p 12.

Remarks: In rabbits, the number of mature follicles in the ovary is known to be relatively constant between individuals, with permanently approximately 8 follicles that may evolve towards ovulation during a 14–18-day cycle (Kranzfelder et al., 1984; doi: 10.1007/BF00219879). Moreover, in a previous study, we demonstrated that diabetic environment in a rabbit model led to a reduction in blastocyst numbers indicating a decrease in fertility in diabetic females (Rousseau-Ralliard et al., 2019; doi: 10.1016/j.mce.2018.10.010.), as observed previously Ramin et al. (Ramin et al., 2010; doi: 10.1210/en.2010-0187) in agreement with data reported in humans.

9.b and 9.c- The higher number of atretic follicles suggests a possible increase in apoptotic mechanisms during folliculogenesis. Unfortunately, this hypothesis has not been investigated using TUNEL in F1 adults. As for F0 H does, the explanations for the increase in atretic follicles might be related to estradiol concentrations known for its anti-apoptotic role [51] but the hormonal profile has not been investigated at this time, or to circulating blood cholesterol since follicular atresia has been observed in rat fed a high cholesterol diet [52]. Moreover, a depletion of antioxidant enzymes associated with an increase in lipid peroxidation and apoptotic markers has been described in this model but also alterations in female serum hormones [52]

This discussion has been added, highlighted in blue, in the part concerning females in « 7.2 Gonadal phenotype » line 513 p12.

  1. Is anything known about the endocrine profiles in the mother and offspring of this model?  Please include if available and discuss as a potential underlying mechanism for the metabolic, cardiovascular and reproductive outcomes.

Results related to endocrine profiles were available for F0 females and have been added and discussed (line 187 and 191 p4), but endocrine profiles were already mentioned in F1 females and males at adulthood (paragraph 7.2 p 12). Elements of discussion supported by bibliographical references from other authors have been added throughout the presentation of the effects and appear highlighted in blue in the text.

- Line 184 p 4, results related to endocrine profiles were supplemented by « In terms of reproduction, the LH (Luteinizing hormone) surge in response to GnRH (Gonadotropin hormone-releasing hormone) challenge was significantly higher in treated females compared to controls at 13 weeks of age, suggested a potential precocious puberty.

- Line 189 p4, results have been added “At 18 weeks of age, no difference of LH was found anymore between the two groups, nor leptin or progesterone whereas estradiol was significantly lower in H compared to C group.»

- Line 187 and 191 p4 , a discussion of F0 maternal phenotype has been added in highlighting blue in the section 4 about “Phenotype of Female Does (F0)”

« An early onset of puberty and of high LH pulse frequency has also been demonstrated in obese rats [49]. A fat-related signal was shown to facilitate the activation of hypothalamic GnRH release and advance the onset of puberty [49]. »

«  Decreased circulating estradiol concentrations were also observed in an obese rat model [50]. Furthermore, the anti-apoptotic role of estradiol could explain the increase of follicular atresia [51] as well as an effect of plasma cholesterol, since a follicular atresia has been reported in a rat model fed with a high cholesterol diet [52]. »

- Moreover, in the general conclusion p16, to summarize F0 phenotype this sentence has been added. “We showed that a diet high in fats (high linoleic acid) and cholesterol induced dyslipidemia and increased fat mass, but without overweight. Before mating, these fatty F0 H females also exhibited aortic atherosclerotic lesions and altered follicular growth and hormonal responses. “ and replaced the sentence “We showed that a high-fat (high linoleic acid) high cholesterol diet induces dyslipidemia, aortic atherosclerotic lesions and increased follicular atresia in F0 females. “

- (line 507 p 12) discussion about endocrine profiles of F1 females on gonadal phenotype were added (highlighted blue text).

« Adult F1 offspring fed the H diet and the ones born to dams fed the H diet had a significantly decreased ovarian area and increased follicular atresia, suggesting that ovarian function is sensitive to prenatal and postnatal H diet without affecting fertility. These data are in agreement with previous studies related to the administration of a high-fat diet. Indeed, a high-fat diet was shown to reduce the ovarian surface area and to increase ovarian atresia in mice, rats and rabbits, impairing follicular reserve [36,79,80]. The higher number of atretic follicles suggests a possible increase in apoptotic mechanisms during folliculogenesis. Unfortunately, this hypothesis has not been investigated using TUNEL in F1 adults. As for F0 H does, the explanations for the increase in atretic follicles might be related to estradiol concentrations known for its anti-apoptotic role [51] but the hormonal profile has not been investigated at this time, or to circulating blood cholesterol since follicular atresia has been observed in rat fed a high cholesterol diet [52]. Moreover, a depletion of antioxidant enzymes associated with an increase in lipid peroxidation and apoptotic markers has been described in this model but also alterations in female serum hormones [52]. The molecular analysis of the ovarian expression of genes involved in ovarian development and oxidative stress was not significantly different between the groups. Nevertheless, the positive correlation between cholesterol and triglycerides and the number of apoptotic follicles, are in agreement with previous data highlighting that cholesterol abundance promote reactive oxygen species and exacerbate apoptotic cell death in the ovarian cell [81]. More recent studies have shown that a high-fat maternal diet affects the expression of genes related to follicle growth in offspring, such as AAT, AFP, and GDF-9, reducing the number of

follicles and alters the development of follicles [80]. Additionally, a high-fat maternal diet also affects ovarian health by inducing ovarian oxidative stress and cellular apoptosis, which collectively can impair the reproductive potential of female offspring. »

- Line 533 p 13, the beginning of the paragraph related to F1 male has been completed by this sentence “For F1 male gonadal phenotype, F1 males were exposed to H or C diet throughout their in utero life until weaning, leading to C-C and H-H groups then, males received a C diet until At 37 weeks of age. »

- Line 541 p 13, a discussion about F1 males on gonadal phenotype was added (highlighted blue text).

« Male gonadal development starts in utero, thus nutritional stress during this period may challenge testicular and epididymis development. As the testes develop slowly from birth to 5 weeks of age in the rabbit, the maternal H diet throughout the gestation and the lactation period could affect the gonadal development as shown in obese rabbits [82]. Maternal overnutrition in sheep has also been associated with alterations in reproductive functions of male offspring, such as lower plasma testosterone concentrations and a reduced testicular volume [83]. So, the maternaldyslipidemia could challenge the concentration of testosterone required to Sertoli cell proliferation and testis development [84]. Moreover, a decrease of testosterone production by Leydig cells could contribute to the low testosterone concentration as observed in obese men [85]. In addition to hormone effects, other mechanisms such as oxidative stress and epigenetic changes could involve in fetal programming and also impair reproductive functions in offspring [86-88] however no effect was observed on sperm concentration, sperm DNA integrity and sperm membrane in our study. »

- Metabolic, cardiovascular and reproductive outcomes have been discussed, including potential mechanisms if described in the literature.

See sentences highlighted in blue throughout the manuscript: line 177p 4, line 187 p4, line 191 p4, line 232 p 6, line 267 p 6, line 312 p8, line 320 p8, line 325 p8, line 334 p8, line 360 p9n line 411 p10, line 440 p11, line 468 p12, line 507 p12, line 542 p12, line 590 p14 and line 642 p 16

  1. Wherever relevant, include more discussion of the findings (rather than detailed results, more about what they mean for the mother, placenta and offspring overall), potential mechanisms responsible, and how the findings compare to those from similar models of maternal high-fat feeding.

- Metabolic, cardiovascular and reproductive outcomes have been discussed, including similar models of maternal high-fat feeding and potential mechanisms if described in the literature.

See sentences highlighted in blue throughout the manuscript: line 54 p2, line 79 p2, line 177p 4, line 187 p4, line 191 p4, line 232 p 6, line 267 p 6, line 312 p8, line 320 p8, line 325 p8, line 334 p8, line 360 p9n line 411 p10, line 440 p11, line 468 p12, line 507 p12, line 542 p12, line 590 p14 and line 642 p 16

Comments on the Quality of English Language

There are just a few grammatical and spelling errors to correct.  For example, line 75 'temporary', line 171 'concentrations' given twice.

- Line 75, now Line 85 p 2 of the revised manuscript : « temporay » has been corrected by « temporary »

- Line 171, now Line 203 p 5 of the revised manuscript  , « concentrations » has been removed

- Line 86, now line 97 p2 of the revised manuscript  « work » has been replaced by « review »

Reviewer 3 Report

In this review, the authors summarized their 15 years of research about the effects of a high-fat maternal diet, mimicking the over-consumption of lipids in European women, on maternal phenotype and offspring, at the embryonic, neonatal, weaning and adult stages, considering sex-specific effects while trying to discriminate the window or windows of vulnerability related to this exposure, in a rabbit model. This review article is well written and organized and can be published in the present form.

Only a few minor concerns:

Line 37: What is LA standing for?

Fig. 1, 2 and 3: What are s (10s), dpc, IUGR standing for?

Line 165: Figure1, a typo;

Line 379: fetuses. (Figure 4). , a typo;  

Line 371: 8.1. “Feto-Placental Phenotype [37]”; The cited reference is not needed for the subheading and is inconsistent with line 373 ref 32. Same condition is found in line 381.

Author Response

Thank you very much for taking the time to review this manuscript. Please find the detailed responses below and the corresponding revisions indicated in red with highlighted in blue in the case of a discussion in the re-submitted files

- Line 37: What is LA standing for?

LA has been defined as linoleic acid line 37 P1

- Fig. 1, 2 and 3: What are s (10s), dpc, IUGR standing for?

« s » has been replaced by w corresponding to weeks on Figure 1 and 3

d.p.c. has been defined in the legend as day post-coitum (Figure 3). p.c. in the title of the legend has beenreplaced by post-coitum and a dot has been added to d.p.c. on the figure 3

IUGR is defined as intrauterine growth restriction on Figure 2

- Line 165: Figure1, a typo;

Figure1 has been changed in Figure 1 line 197

- Line 379: fetuses. (Figure 4). , a typo;

The dot before (Figure 4) has been removed line 596 p 15

- Line 371: 8.1. “Feto-Placental Phenotype [37]”; The cited reference is not needed for the subheading and is inconsistent with line 373 ref 32. Same condition is found in line 381.

Line 371, the reference in the subheading has been removed (now line 585 p14) and replaced the reference 32 line 587 p14

Line 381, the reference in the subheading has been removed and placed line 599 p15

Round 2

Reviewer 1 Report

The authors replied in a satisfying way to my requests. The paper is now more clear and readable.

The authors replied in a satisfying way to my requests. The paper is now more clear and readable.